# Combined COX-2/PPARγ Expression as Independent Negative Prognosticator for Vulvar Cancer Patients

**DOI:** 10.3390/diagnostics11030491

**Published:** 2021-03-10

**Authors:** Nadine Ansorge, Christian Dannecker, Udo Jeschke, Elisa Schmoeckel, Doris Mayr, Helene H. Heidegger, Aurelia Vattai, Maximiliane Burgmann, Bastian Czogalla, Sven Mahner, Sophie Fuerst

**Affiliations:** 1Department of Obstetrics and Gynecology, University Hospital, LMU Munich, Marchionini Str. 15, 81377 Munich, Germany; nadine.ansorge@uk-augsburg.de (N.A.); helene.heidegger@med.uni-muenchen.de (H.H.H.); aurelia.vattai@med.uni-muenchen.de (A.V.); maximiliane.burgmann@med.uni-muenchen.de (M.B.); bastian.czogalla@med.uni-muenchen.de (B.C.); sven.mahner@med.uni-muenchen.de (S.M.); sophie.fuerst@med.uni-muenchen.de (S.F.); 2Department of Obstetrics and Gynecology, University Hospital Augsburg, Stenglinstr. 2, 86156 Augsburg, Germany; christian.dannecker@uk-augsburg.de; 3Department of Pathology, LMU Munich, Thalkirchner Str. 37, 80337 Munich, Germany; elisa.schmoeckel@med.uni-muenchen.de (E.S.); doris.mayr@med.uni-muenchen.de (D.M.)

**Keywords:** COX-2, PPARγ, vulvar cancer, survival

## Abstract

Vulvar cancer incidence numbers have been rising steadily over the past decades. Especially the number of young patients with vulvar cancer increased recently. Therefore, the need to identify new prognostic factors for vulvar carcinoma is more apparent. Cyclooxygenase-2 (COX-2) has long been an object of scientific interest in the context of carcinogenesis. This enzyme is involved in prostaglandin synthesis and the latter binds to nuclear receptors like PPARγ. Therefore, the aim of this study was to investigate COX-2- and PPARγ- expression in tissues of vulvar carcinomas and to analyze their relevance as prognostic factors. The cytoplasmatic expression of COX-2 as well as PPARγ is associated with a significantly reduced survival, whereas nuclear expression of PPARγ results in a better survival. Especially the combined expression of both COX-2 and PPARγ in the cytoplasm is an independent negative prognosticator for vulvar cancer patients.

## 1. Introduction

In 2018, more than 15,000 women worldwide died of vulvar cancer. However, with a worldwide incidence of 44,235 new cases, the number has been rising steadily over the past decades [1]. In addition, there is a continuing increase in new cases in young females [2,3].

A total of 90% of vulvar carcinomas are squamous cell carcinomas (VSCC). Unkeratinized squamous cell carcinomas are often human papilloma virus (HPV)-associated and mainly affect postmenopausal women [4,5]. Keratinized squamous cell carcinomas, on the other hand, are mostly due to a chronic genital disease such as lichen sclerosis [6,7]. In addition to HPV, other risk factors are associated with the development of vulvar carcinoma: immunosuppression, smoking [8] and sexually transmitted diseases such as herpes simplex virus 2 infections [5] are associated with an increased risk.

With regard to therapy, surgical interventions are predominantly used which end often in a vulvectomy if the findings are extensive. The consequences of such a serious and extensive surgery have been rarely studied to this date. Restrictions in sexual behavior, micturition problems or even psychological effects impairing the quality of life are late effects of this radical form of therapy [9,10]. Regarding prevention, the HPV vaccination, for instance, was seen as a great beacon of hope in the fight against HPV-associated tumors such as cervical, anal, and vulvar cancer [11,12]. The EURO vaccination meeting 2016 listed Belgium as a top performer with a vaccination rate of 84%, while in Germany the vaccination rate reached critically 31% in 2015 [13].

Based on the low vaccination rate in Germany it can be assumed that the need for newly found prognostic factors for vulvar carcinoma is even more apparent. In addition, we find an increasing number of new cases, younger patients, radical therapy, and no comprehensive prevention due to the low vaccination rates, at least in this country.

Cyclooxygenase-2 (COX-2) has long been an object of scientific interest in the context of carcinogenesis. In contrast to constitutive housekeeping enzyme COX-1, COX-2 is as a known enzyme of inflammation inductively expressed [14,15]. Exceptions regarding constitutive COX-2 expression are tissues of the brain, kidneys, testes, and tracheal epithelium [16,17]. The induction of the COX-2 enzyme is triggered by cell damage or inflammation through the release of various factors such as growth factors like epidermal growth factor (EGF) [18], prostaglandins, or chemokines like TNF-γ [19]. It is assumed that products of COX-2 like prostaglandin E2 (PGE2) have a decisive influence on the development of tumors, e.g., in angiogenesis [20,21].

The nuclear receptor superfamily for steroids, hormones, vitamin D, and retinoid is formed by isoforms like PPARα, PPARß, and PPARγ (peroxisome proliferator activated receptor gamma). In general, PPARs act as ligand-dependent transcription factors that bind to specific DNA sequences, the PPREs (PPAR response elements). After heterodimerization with retinoid X-receptor (RXR) a regulatory effect on transcription can occur. Ligands altering the conformation of the receptors lead to co-activation or -repression [22,23,24,25,26]. PPARγ has a proven influence on the regulation of insulin sensitivity and glucose metabolism. Based on this knowledge, the PPARγ agonist from the group of thiazolidinediones made its way into the pharmacological therapy of type 2 diabetes mellitus [27]. An interesting observation of PPARγ and its activators like prostaglandin J2 is the effect on cell differentiation [28,29], cell proliferation and apoptosis induction [30,31]. The expression and related antiproliferative property have been demonstrated in some carcinomas [32,33,34,35].

This study investigates COX-2- and PPARγ- expression in tissues of vulvar carcinomas and their relevance as prognostic factors.

## 2. Materials and Methods

### 2.1. Clinical Data and Tissue Collection

177 patients with vulvar carcinoma primarily diagnosed in the period from 1990 to 2008 were included in this study. The entire patient group was treated at the department of Gynecology and Obstetrics of the Ludwig-Maximilians-University in Munich, Germany. Surgically obtained tissue samples were histopathologically processed and specified. All follow-up and survival data were provided by the tumor register of Munich.

For immunohistochemical staining, 157 of the 177 samples were available. During the evaluation, a further 16 tissue samples were excluded, as the incisions did not contain a tumor, but only precancerous stages of the carcinoma. Therefore, in the end a collective of 141 slides was assessed.

Median age of the investigated collective was 70 years, ranging from 20 to 96 years, with 72 of the 141 patients younger than 70 years (=51.8%% of the collective). All relevant clinic-pathologic parameters are listed in Table 1 below.

### 2.2. Ethical Approval

All patients’ data were completely anonymized, and the study performance was carried out according to the standards set in the Declaration of Helsinki 1975. The examined tissues were residual material that had been collected in first instance for histopathological diagnostic procedures. The actual study was approved in writing by the Ethics Committee of the Ludwig-Maximilians-University, Munich, Germany (approval number 367-16, 29 December 2016). Authors were blinded for clinical information during experimental analysis.

### 2.3. Immunohistochemistry

After formalin-fixing and paraffin-embedding, all samples were cut to 4 µm from paraffin block. They were mounted on SuperFrost Plus microscope slides (Menzel Glaeser, Braunschweig, Germany). For deparaffinizing tissue patterns were processed with xylol for 20 min and washed by 100% ethanol. All slides were prepared with 3% hydrogen peroxide diluted in methanol for 20 min to stop activity of endogenous peroxidase. Afterwards rehydration took place in a descending alcohol series (100%, 70%, 50%) and were washed with distilled water. The samples were heated with citric acid buffer in a pressure cooker to uncover epitopes of antigens. Furthermore, slides were washed two times with phosphate buffered saline (PBS). Zytochem-Plus HRP Polymer-kit (Zytomed, Berlin, Germany) was utilized for blocking and antibody staining. After saturating electrostatic charges in tissue with blocking solution for 5 min, either the polyclonal rabbit IgG anti- COX-2 antibody (Sigma, St. Louis, MO, USA, SAB4502491) or the polyclonal rabbit IgG anti- PPARγ antibody (abcam, Cambridge, Great Britain, ab59256) were applied on tissue specimens. Anti-COX-2- antibody was diluted at a ratio of 1:400 and anti- PPARγ -antibody at a ratio of 1:100. The incubation time of both antibodies amounts to 16 h at 4 °C in humid chamber. Slides were incubated by post-block reagent for 20 min and thereafter by HRP-Polymer for 30 min at room temperature in a humid chamber. After each application with the antibody, post-block and HRP-Polymer the samples were washed two times with PBS. 3,3′-Diaminobenzidine (Dako, Hamburg, Germany) catalyzed the peroxidase substrate staining so that the color precipitation is detectable with a light microscope. Finally, slides were counterstained with hemalum, again washed by 100% ethanol and covered with glass. As positive control, both antibodies were stained in placenta tissue for validating the staining method.

Under use of the semi quantitative immunoreactive score (IRS) by Remmele and Stegner [36], tissue patterns were evaluated with the light microscope (Leitz, Wetzlar, Germany). For this purpose, the intensity score and the percentage score in the tumor tissue were formed. The intensity score is divided into 0 = no, 1 = weak, 2 = moderate, 3 = strong; the percentage score is also categorized into 0 = no staining, 1 ≤ 10%, 2 = 11% to 50%, 3 = 51% to 80%, 4 ≥ 81%. IRS score is formed by product of both scores (intensity score x percentage score). The antibodies showed expressions in cytoplasm and in nucleus, so both expression templates were examined independently by IRS. Patients’ data were correlated by IRS and by its two IRS-forming factors of staining intensity and percentage of positively stained cells.

### 2.4. Statistical Analysis

For statistical analysis, the SPSS Statistic version 25 (IBM Corp., Armonk, NY, USA) was used. The non-parametric Kruskal-Wallis test was used to compare between and among groups. Correlation analyses were performed using the Spearman rank correlation coefficient. Kaplan-Meier curves were generated using collected survival data, differences between these curves were tested by the log-rank test. The level of statistical significance was accepted at *p* ≤ 0.05 and all test were two-sided.

## 3. Results

### 3.1. COX-2 as Predictor for Grading and for Overall Survival

In the patient group, 91.3% of the stained samples were positive for COX-2 in the cytoplasm. The immunohistochemical evaluation showed a positive correlation of COX-2 amount in the cytoplasm of the tumor tissue to the respective degree of differentiation (grading) (Spearman-Rho, * *p* = 0.003, Figure 1).

There is a distinct statistical correlation between the increase in COX-2 expression per immunoreactive score (IRS) and the grading (Spearman-Rho, * *p* = 0.020, images B-D in Figure 1). Considering tumor stage T, there is also a significant, concurrent relationship to the proportion of COX-2 expression in tumor tissue (Spearman-Rho, * *p* = 0.021). Furthermore, a reduction in overall survival was demonstrated for the group of patients who had an IRS value > 3 in the cytoplasm in COX-2 staining (* *p* = 0.003, Figure 2).

A total of 49.3% of the tumor samples have an IRS > 3 for COX-2 in the cytoplasm, the remaining 50.7% were below this IRS value. As the Kaplan-Meier illustrates, the 10-year overall survival of patients with an IRS value >3 was 20%, but patients with a lower IRS value lived more than twice as long at 46% (Figure 2). This data shows a median survival advantage of patients with lower IRS values (≤3) compared to patients with higher IRS values (>3) at 40 months (Table 2).

The multivariate analysis revealed that grading (* *p* = 0.004), p16 status (* *p* = 0.001), and COX-2 (* *p* = 0.005) functioned as independent prognostic factors for overall survival. However, tumor stage, nodal status, and FIGO classification did not act as independent prognostic factors (Table 3).

### 3.2. PPARγ as a Negative Prognostic Factor for Disease-Free Survival in Cytoplasm

A total of 78.4% of the tumor samples were positive for PPARγ in the cytoplasm, 21.6% showed no cytoplasmic staining. The intensity of PPARγ expression in the cytoplasm showed a positive correlation with the progression status (* *p* = 0.008) and the development of a local recurrence (* *p* = 0.016, all Spearman-Rho test). The Kaplan-Meier curve showed a significantly worse disease-free survival for patients with a PPARγ expression ≥ 2 in cytoplasm of the tumor tissue than for those whose IRS value is below 2 (* *p* = 0.036, Figure 3).

After 10 years, 62% of tumor patients with an IRS value less than 2 lived disease-free. However, patients with a higher IRS value had a shorter disease-free survival (51% after 10 years). The median survival data showed an absolute survival advantage of 63 months for patients with IRS values below 2 (Table 4).

In 23.0% of the patients an IRS value ≥ 2 for PPARγ in the cytoplasm was found. Finally, the Cox regression analysis concluded that PPARγ was not an independent factor for disease-free survival in vulvar cancer (Table 5, *p* = 0.626). The independent prognostic factor in disease-free survival, though, was grading (Table 5, * *p* = 0.009).

### 3.3. Nuclear PPARγ as a Positive Prognostic Factor for Overall Survival

Within our group of patients, a nuclear expression for PPARγ could be detected with the exception of only 2 cases. In total, 98.6% of the investigated patient group showed a positive expression pattern in the nucleus for PPARγ. The survival curve showed that the nuclear expression of PPARγ had a positive effect on overall survival at values ≥ 2 (Figure 4, image A, * *p* = 0.019). In comparison, the expression of PPARγ in the cytoplasm is associated with a negative trend in overall survival (Figure 4, image B, *p* = 0.053).

### 3.4. Correlation between PPARγ and p16 Status

PPARγ (nuclear expression) shows a clear statistical negative correlation to the p16 status (* *p* = 0.004, Spearman-Rho test). Moreover, the Spearman-Rho test proved that the nodal status also has a negative correlation to the IRS value of PPARγ in the nucleus (* *p* = 0.017) which indeed underlines the survival advantage with higher nuclear expression of PPARγ in the nucleus.

### 3.5. Combined COX-2/PPARγ Expression as an Independent Prognostic Factor for Overall Survival

In addition to the individual studies of COX-2 and PPARγ regarding survival, a significantly stronger influence of both factors together was observed. This resulted in the association of a low to absent expression of one or both factors with the longest overall (** *p* < 0.001, Figure 5, image A) and disease-free survival (* *p* = 0.006, Figure 5, image B).

Low expression was determined below an IRS value of 3 in the cytoplasm. In comparison, 10-year survival is more than twice as long in patients with low IRS for COX-2 and/or PPARγ. Disease-free survival is also extended from 49% to 70% with values of IRS ≤ 3. Patients with an IRS value ≥ 3 live a median of 126 months, whereas patients with lower score values live only 48 months. The disease-free survival is similar: median disease-free survival is 205 months for patients with IRS values ≥ 3, while patients with lower IRS values spend 86 months less disease-free. A total of 57.7% of tumor patients were positive for COX-2 and/or PPARγ with a cytoplasmic IRS value >3; 42.3% did not have a value > 3 for either factor. The independence of COX-2 and PPARγ as a predictive factor for overall survival from other clinical pathological factors was tested by a cox regression analysis (* *p* = 0.001, Table 6) in comparison to tumor stage, nodal state, grading, FIGO- classification, and p16-state.

## 4. Discussion

Our study proved that the expression of COX-2 and PPARγ and their combination in the tissue of the vulvar carcinoma has a strong impact on survival (Figure 2, Figure 3, Figure 4 and Figure 5).

So far, there are only a few studies describing prognostic factors in vulvar carcinoma. The multicenter AGO-CaRE study [37] reported lymph node metastases as a decisive prognostic factor for patients with vulvar carcinoma [38]. Concerning potential prognostic markers showing significant relation to survival in vulvar cancer, only small studies addressed amongst others on p16 [39], p53 [40], ERβ [41], c-KIT [42], p14ARF [43].

COX-2 plays a decisive role in carcinogenesis. Acting COX-2 products, prostanoids, appear to be linked to the development and progression of a tumor disease. Processes including angiogenesis, invasion, apoptosis inhibition, growth, and aggressiveness of the tumor seem to depend strongly on COX-2 and its products [44,45,46]. Various cross-links to signaling pathways via NF-kB [47], Wnt/ß-catenin [48], PI3K/AKT [49], or activations of anti-apoptotic Bcl-2 [50] are established by COX-2. The NF-kB pathway regulates the expression of COX-2. As it has already been investigated in several studies, PPARγ in its activated form acts in the nucleus as an inhibitory factor on the transcription factor NF-kB. PPARγ thereby inhibits the expression of COX-2, which can be regarded as one important connection point between the two proteins [47,51,52].

Furthermore, blocking effects of prostaglandin E2 and prostaglandin F2α on the pro-apoptotic PPARγ have been reported [53,54]. However, there are some isolated studies to the contrary that pronounce COX-2 to have anti-tumor properties also [55,56].

Hence, our observation that COX-2 in the cytoplasm is a highly significant independent prognostic factor for the overall survival of our patient population is even more interesting.

Increased COX-2 expression was not only found in a variety of gynecological tumors such as endometrial carcinoma [57], breast carcinoma [58,59,60], ovarian carcinoma [61], or cervical carcinoma [62], but also in a lot of other tumor entities [63,64,65,66,67,68]. Only a few studies exist regarding the expression of COX-2 in tumor tissue of the vulva [69,70].

In the study by Fons et al. [70], COX-2 has already been associated with poorer overall survival in vulvar cancer in a smaller number of cases (*n* = 50), but did not prove to be an independent factor. Comparing results with other tumor entities, a role as an independent prognostic factor was e.g., detected by Mrena et al. [71] in gastric carcinoma. Becker et al. [72] also reported a significant correlation between COX-2 levels in malignant melanoma and overall survival.

Apart from this, COX-2 has a direct relationship with tumor grading and tumor stage in our study. Lee et al. [73] observed an inverse relationship between grading and COX-2 levels in patients with vulvar cancer, where low grading stages had the highest COX-2 expression levels.

Our observation of a positive correlation between COX-2 and tumor stage and/or grading goes along with the results by Sheehan et al. [74] in colon cancer, Miyata et al. [75] in renal cell carcinoma, and Boland et al. [59] in DCIS of breast.

In terms of the prognostic significance of COX-2 in certain tumor entities, however, improved survival was also observed [76,77]. The prognostic significance of PPARγ in vulvar cancer is clearly different depending on its localization of expression. In the nucleus, PPARγ is actively involved in the regulation of gene expression in its role as a transcription factor [23,25]. Nevertheless, we also found that PPARγ can also be stained in the cytoplasm of vulvar carcinoma. Several other tumor tissues have been identified as having such a staining profile [78,79,80].

Due to the fact that PPARγ cannot act in the cytoplasm in its genomic function as a transcription factor, the expression of PPARγ in the cytoplasm is assumed to be associated with the lack of nuclear activity in gene regulation and a non-genomic functioning in cytoplasm. The translocation dynamics between cytoplasm and nucleus are getting in focus of scientific research to an increasing extent. Like other nuclear receptors such as progesterone receptor [81,82], glucocorticoid receptor [83,84], androgen receptor [85], or thyroid receptor [86], the nuclear-cytoplasmic shuttling is an important component of regulating the activity of these receptors. For now, there is no clearly identifiable shuttle protein involved in trafficking of PPARγ and the cytoplasmic function of PPARγ is widely unexplained, but often aim of scientific experiments [87,88,89,90]. Other factors influence the activity regulation of PPARγ like ubiquitination or the influence of natural ligands like PGJ2 or external ligands like thiazolidiones [91]. Additional influences by post-translational modification via phosphorylation by MAPK as well as activity modulation by traditional herbal medicine plants like V.trifolia demonstrate the diversity of possible modulatory pathways of the activation and expression pattern of PPARγ [92,93].

These findings would explain the opposite effect on survival when PPARγ is detected in the nucleus or in the cytoplasm: Nuclear expressed PPARγ is a prognostically favorable factor in overall survival, but detection in the cytoplasm is a prognostically unfavorable factor in disease-free survival and shows an unfavorable trend in overall survival. Shao et al. [94] described an unfavorable overall survival in breast cancer patients showing a high-expression level of PPARγ in cytoplasma. In some tumor entities, a positive prognostic influence of nuclear PPARγ expression on the survival of tumor patients has already been identified, but a difference between expression localization and relation to different patient outcome was not described [95,96]. In the present study, a predictive difference between nuclear expression and cytoplasmic expression of PPARγ in vulvar cancer patients is investigated and described for the first time.

Several in vitro and in vivo studies have suggested that PPARγ is effective as a tumor suppressor. Nicol et al. [97] demonstrated an increased rate of developed neoplasia in mammary, ovary and skin in PPARγ-deficient mice. Sarraf et al. [98] describes loss-of-function mutations of PPARγ in colon cancer tissue as a contributing factor to tumorigenesis. Furthermore, the pro-apoptotic and anti-proliferative effect in tumor cell lines with PPARγ agonists could be proven in some studies [34,99,100,101,102]. The tumor suppressing property of PPARγ remains controversial. In contrast, Lefebvre [103] and Saez et al. [104] showed that ligand-induced activation of PPARγ in mouse experiments resulted in the occurrence of colonic polyps and an increased probability of degeneration.

Controversially discussed prognostic properties of PPARγ may be explained by tissue-specific effects and the concentration of the PPARγ ligands used. The detailed review of Clay et al. [105] showed that experiments with the PPARγ agonist PGJ2 induce carcinoma growth under low-dose conditions, but a decrease in proliferation behavior under high-dose conditions.

In studies of oropharyngeal squamous cell carcinoma, p16 is considered a surrogate marker for HPV positivity. Our research group has already investigated a proven positive effect of p16 on the prognostic outcome of patients with VSCC. p16 positive VSCC revealed a longer overall and progression free survival [106]. It also appears in some other studies that HPV-associated VSCC have a better clinical outcome as Sand et al. [107] reported in a recent review. However, our study revealed a negative correlation of nuclear PPARγ and p16. Therefore, p16 seems not to be involved in PPARγ translocation.

The combination of the cytoplasmic expression of COX-2 and PPARγ showed that both factors together have the strongest predictive power for a negative survival prognosis in overall and disease-free survival. COX-2 and PPARγ together function independent from other clinical pathological parameters as strongly significant prognostic factors for patients with vulvar cancer.

The enzyme COX-2 and the transcription factor PPARγ are interacting. PGE2 inhibits the activity of the pro-apoptotic active transcription factor and in turn underlines the carcinogenic effect of COX-2 [20,21,54]. Studies by Rothwell et al. using COX inhibitors showed impressive results in terms of improving the prognosis and reducing the incidence of colorectal cancer [108].

In addition, there is also an activating link between the two molecules via PGJ2, the natural ligand and agonist of PPARγ. Consequently, the anti-tumor effect of PPARγ is supported [30,32]. These two modes of activity illustrate clearly that the importance does not only lie in the anti-tumor or pro-tumor effects of molecules, but the respective predominance of a characteristic molecule in the individual and tissue-specific context. Factors that influence this balance, for example through shuttling with resulting activation or inactivation, require closer observation.

The limitations of our study are most likely the use of one method only to detect the expression level of COX-2 and PPARγ in the tissue sections. Immunohistochemistry was applied in our single-method approach due to the fact that this technique of advanced histopathological diagnostic is well established and renowned within the respective field of research. This limitation comes along with the retrospective design of the study. Only a highly limited number of patients included in this collective was alive during the examination of the tissue sections so that there would have been the possibility of applying a scan for cells within serum or primary cells.

However, this subjective form of evaluation was objectified by the use of two independent investigators who evaluated the expression pattern blinded.

There was no possibility to overview the process of the embedment of the tissue which leads to the fact that in the analyzed collective only tumorous tissue is accessible. Vulvar tissue patterns of patients with non-malignant vulvar diseases were tested by immunohistochemical expression level for COX-2 and PPARγ for clarifying differences between malignant and non-malignant tissue expression levels (Appendix A).

In addition, only sections containing invasive vulvar carcinoma assessed by an experienced pathologist were evaluated. Furthermore, immunohistochemistry is an integral part of tumor diagnostics, which is still mainly used in gynecology and is highly appreciated as an economical and simple method in clinical routine.

The method of immunohistochemical staining used in this study reflected the expression level but is hardly meaningful regarding the activity of the stained proteins. Thus, conclusions like the possible improvement of the outcome by using COX-2 inhibitors or PPARγ agonists cannot be drawn.

The data of the collective includes no information about the drug status of the patients at the time of tissue collection; therefore, no conclusion can be drawn about a possible intake of a COX inhibitor or PPARγ agonist. However, medication is unlikely to alter the expression pattern, as the drugs only affect the activity and not the expression.

In contrast to other immunohistochemical studies on vulvar cancer, we have a very large collective with real-time data from patients. In 2016, the Robert-Koch-Institute Germany reports regarding the epidemiology of vulvar carcinoma a medium age of 73 years, which is manifested in the highest burden of disease within the group of women over 70 [109]. In this case our collective represents an approximation of the frequency of this disease within the age groups. The aim of our study is to find a prognostic factor for all vulvar carcinoma patients. Due to the rarity of this type of carcinoma, especially in comparison to other types e.g., breast cancer, the sample size of our collective is unprecedented in common literature.

## 5. Conclusions

After demonstrating that COX-2 and PPARγ are prognostic factors for overall and disease-free survival in vulvar cancer, research should continue on further possible pathways linking COX-2 and PPARγ. Both molecules should be perceived as potential targets in the context of vulvar cancer therapy. Further in vitro experiments as well as a transfer into prospective clinical models must be reconsidered and their urgent necessity recognized. This would be a great opportunity for a patient collective that has so far received little attention, with the prospect of less invasive, more biomolecular, and individualized therapeutic approaches that could not only ensure survival but also improve the quality of life.

## Figures and Tables

**Figure 1 diagnostics-11-00491-f001:**
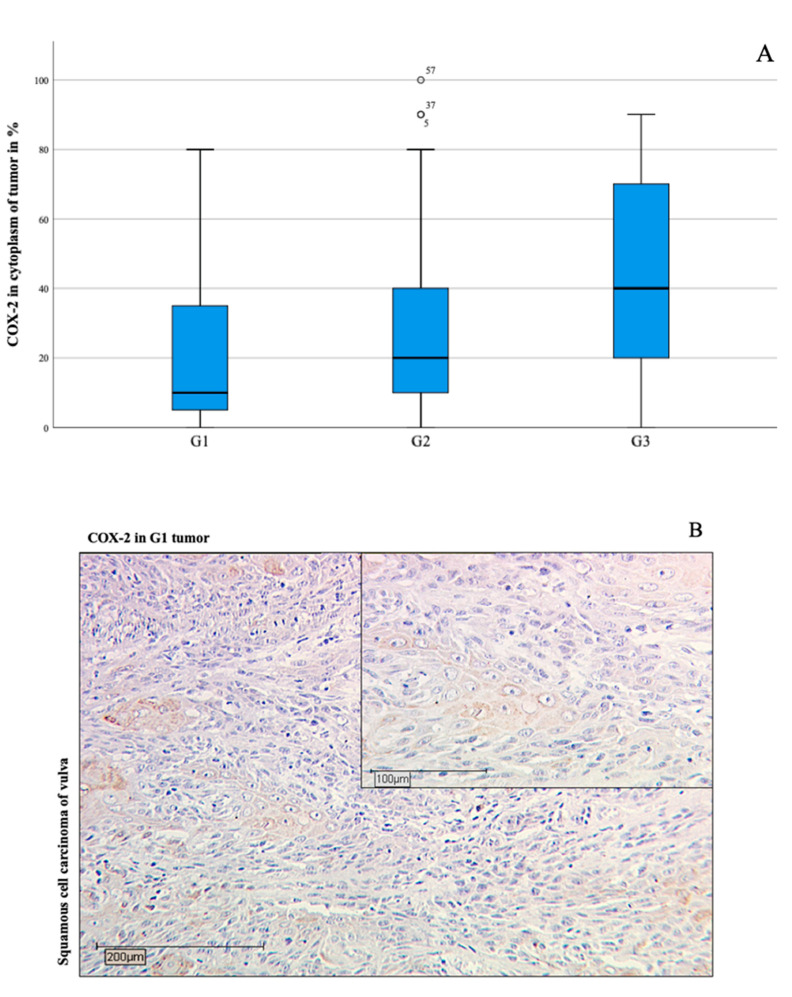
Boxplots (**A**) presenting positive correlation (* *p* = 0.001 in Spearman-Rho) between COX-2 positive tissue amount and the individual degree of tumor grading (G1: well-differentiated for (**B**), G2: moderately-differentiated for (**C**), G3: poorly-differentiated for (**D**)). The boxplots indicate mild outliers, which are marked with circles. These outliers show an interquartile distance to the third quartile of values that is less than three times higher than the third quartile of values. The numbers on the circles denote the cases (case numbers 5, 37, 57) in concern. Immunohistochemistry staining of cytoplasmic COX-2 (10× and 25× magnification) showing correlation to Grading 1–3 with increase of COX-2 intensity in vulvar cancer (**B**–**D**). The medians of the percentage COX-2 expression shown in the boxplots of the individual grading categories are represented in the immunohistochemical images of (**B**–**D**) (amount 10% in (**B**), amount 20% in (**C**), and amount 40% in (**D**)).

**Figure 2 diagnostics-11-00491-f002:**
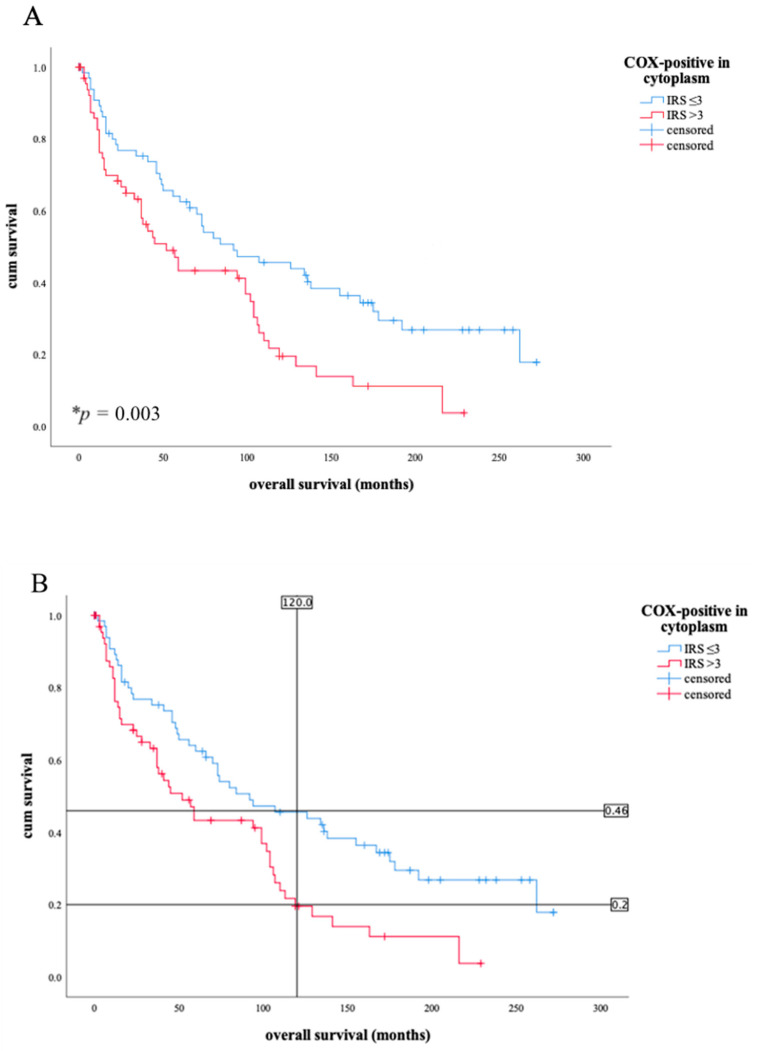
As the Kaplan-Meier curve shows, patients with cytoplasmatic COX-2 expression according to immunoreactive score (IRS) > 3 survive for a shorter time period than patients with a lower IRS ((**A**), * *p* = 0.003). The blue line shows the survival curve of patients with COX-2 expression of IRS level ≤ 3, the red line shows the survival curve of patients with COX-2 expression IRS level above. The 10 year survival ((**B**), 120 months signed with vertical line) of a patient with IRS > 3 is about half as high as that of a patient with lower IRS. The horizontal lines in (**B**) illustrate the points of intersection on the Kaplan-Meier curves: IRS values above 3 show that 20% of patients live after 10 years; IRS values below 3 demonstrate that 46% of patients survive after the same time.

**Figure 3 diagnostics-11-00491-f003:**
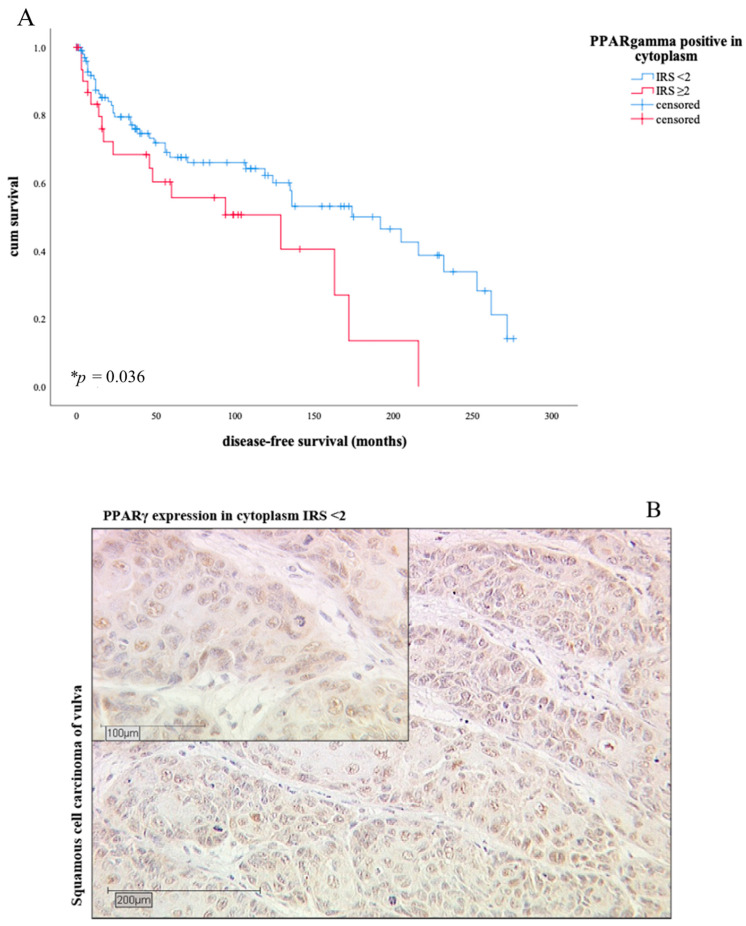
As the Kaplan-Meier curve (**A**) shows, patients with cytoplasmic PPARγ expression according to IRS ≥ 2 survive a shorter time than patients with a lower IRS (* *p* = 0.036). The blue line shows the survival curve of patients with PPARγ expression of IRS level under 2, the red line shows the survival curve of patients with PPARγ expression IRS level ≥ 2. (**B**) shows an example of low expression level of PPARγ in cytoplasm (IRS < 2), (**C**) represent a high expression level of PPARγ (IRS ≥ 2) in cytoplasm, in contrast.

**Figure 4 diagnostics-11-00491-f004:**
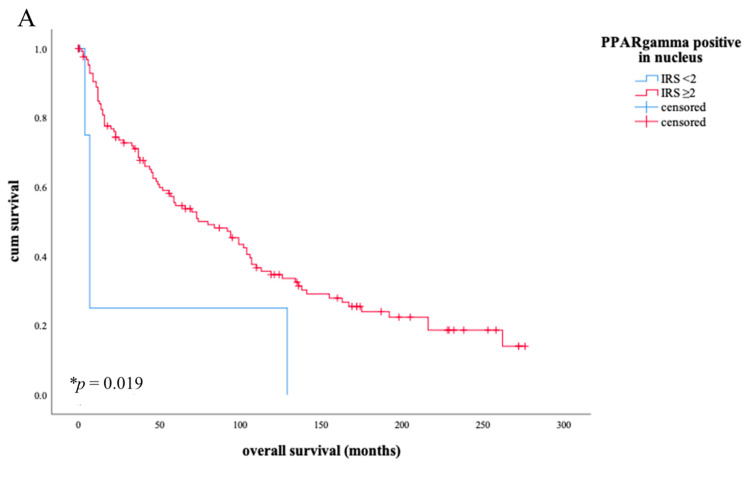
As the Kaplan-Meier curve illustrates, an IRS value ≥2 of the nuclear PPARγ expression is related to a longer overall survival than in patients with lower values ((**A**), * *p* = 0.019). Here, a prognostic survival advantage is shown in contrast to the contrary trend in the Kaplan-Meier curve in Figure 4B. The blue line in Figure 4A shows the survival curve of patients with nuclear PPARγ expression of IRS level under 2, the red line shows the survival curve of patients with nuclear PPARγ expression IRS level ≥2. (**B**) reveals an opposite trend in the overall survival curve as soon as PPARγ expression is detected in the cytoplasm rather than in the nucleus (**B**), *p* = 0.053). The blue line in (**B**) shows the survival curve of patients with PPARγ expression in cytoplasm of IRS level under 2, the red line shows the survival curve of patients with PPARγ expression in cytoplasm of IRS level ≥ 2.

**Figure 5 diagnostics-11-00491-f005:**
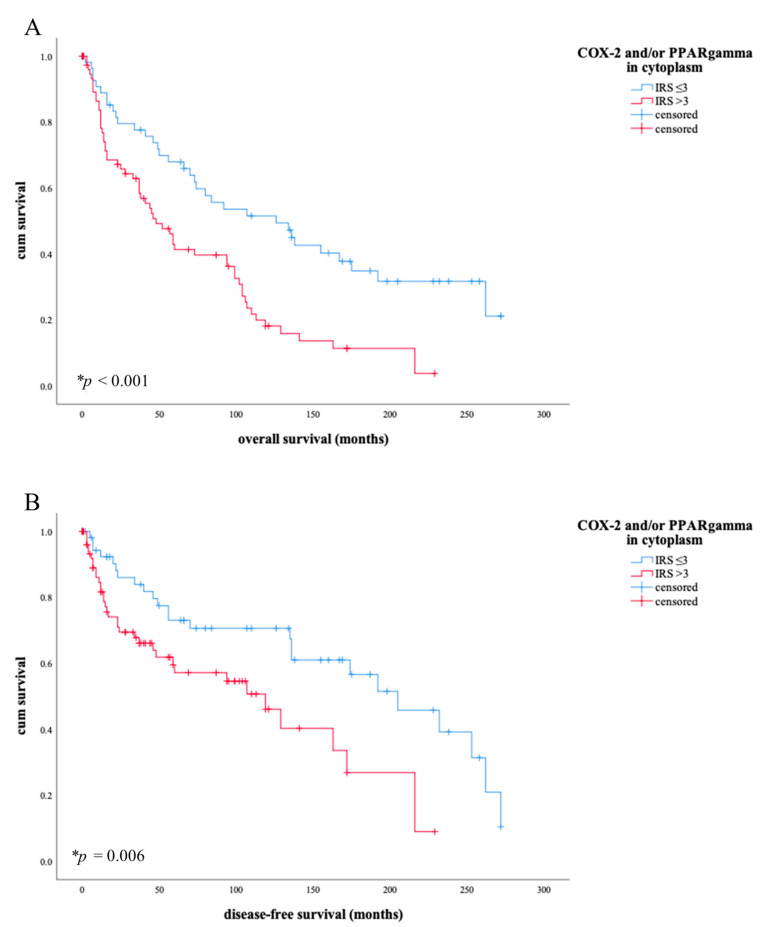
When COX-2/PPARγ is expressed with IRS in cytoplasm >3, a shorter overall survival ((**A**), * *p* < 0.001) but also with regard to disease-free survival (**B**), * *p* = 0.006) can be derived. The blue line in (**A**) shows the overall survival of patients with COX-2/PPARγ expression in cytoplasm of IRS level ≤ 3, the red line shows the survival curve of patients with COX-2/PPARγ expression in cytoplasm of IRS level above 3. The blue line in (**B**) shows disease-free survival of patients with COX-2/PPARγexpression in cytoplasm of IRS level ≤ 3, the red line shows the survival curve of patients with PPARγexpression in cytoplasm of IRS level above 3. After 10 years, more than twice as many patients (0.5) live with lower IRS values than patients with expression of one or both factors (0.21) after IRS > 3. The situation is similar in disease-free survival: lower expression of COX-2 and/or PPARγ shows longer disease-free survival (0.7) than with higher expression of one or both factors (0.49).

**Table 1 diagnostics-11-00491-t001:** Clinicopathological Parameters of Vulvar Carcinoma Patients’ Collective.

Clinicopathologic Parameters	*n*	Percentage (%)
Histology		
keratinizing	134	95
wartv/basaloid	7	5
Tumor size		
T1	51	36.2
T2	74	52.5
T3	9	6.4
missing	7	5
Nodal status		
N0	60	42.6
N1	31	22
N2	8	5.7
missing	42	29.8
Metastasis		
missing	141	100
FIGO		
I	45	31.9
II	45	31.9
III	36	25.5
IV	9	6.4
missing	6	4.3
Grading		
G1	24	17
G2	87	61.7
G3	29	20.6
missing	1	0.7
p16 status		
positive	34	24.1
negative	57	40.4
missing	50	35.5
Progression status		
positive	61	43.3
negative	79	56
missing	1	0.7
Local recurrence status		
positive	35	24.8
negative	105	74.5
missing	1	0.7

**Table 2 diagnostics-11-00491-t002:** There is a clear difference in overall survival for patients with IRS values for COX-2 expression in the cytoplasm above 3. Patients with IRS values above 3 live with a median of 52 months, whereas patients with lower IRS values survive 92 months. As the table shows, the survival of patients of both groups differs by 40 months, i.e., more than three years. Even in the total group a survival difference is recorded: patients live a total of 73 months, but still lose 21 months of life with higher IRS values.

Median for Overall SuMedian for Overall Survival Time (Months)			
COX-2 IRS Value in Cytoplasm	Estimate	Lower 95% Confidence Interval	Upper 95% Confidence Interval
IRS ≤ 3	92.000	36.414	147.586
IRS > 3	52.000	31.292	72.708
Overall	73.000	44.681	101.319

**Table 3 diagnostics-11-00491-t003:** Cox regression of clinical-pathological variables regarding overall survival in vulvar carcinoma patients.

Variable	Significance	Hazard Ratio of Exp (B)	Lower 95% Confidence Interval of Exp (B)	Upper 95% Confidence Interval of Exp (B)
COX-2 in cytoplasm	0.005	2.187	1.267	3.776
pT	0.488	1.275	0.642	2.535
pN	0.112	1.005	0.999	1.012
Grading	0.004	1.874	1.222	2.873
FIGO	0.199	1.336	0.858	2.081
p16 status	0.001	0.362	0.196	0.671

COX-2 cytoplasm = expression of COX-2 in cytoplasm with IRS > 3, pT = tumor stage, pN = nodal stage, FIGO = Classification of the International Federation of Gynecology and Obstetrics.

**Table 4 diagnostics-11-00491-t004:** The table shows that an expression of PPARγ in the cytoplasm from IRS values of 2 upwards is a survival disadvantage in disease-free survival. The tumor patients with IRS values of 2 and above live disease-free for a median of 129 months; Patients with lower IRS values, however, live 192 months and thus much longer. Overall the patients live 163 months. The difference between both IRS groups (from 2 or below) is a disease-free survival difference of 63 months, i.e., more than 5 years.

Medians for Disease-Free Survival Time (Months)			
PPARγ IRS Value in Cytoplasm	Estimate	Lower 95% Confidence Interval	Upper 95% Confidence Interval
IRS < 2	192.000	127.692	256.308
IRS ≥ 2	129.000	20.907	237.093
Overall	163.000	203.332	203.332

**Table 5 diagnostics-11-00491-t005:** Cox regression of clinical-pathological variables regarding overall survival in vulvar carcinoma patients.

Variable	Significance	Hazard Ratio of Exp (B)	Lower 95% Confidence Interval of Exp (B)	Upper 95% Confidence Interval of Exp (B)
PPARγ in cytoplasm	0.626	1.193	0.588	2.418
pT	0.336	1.496	0.658	3.400
pN	0.403	0.996	0.985	1.006
Grading	0.009	2.016	1.190	3.413
FIGO	0.259	1.342	0.805	2.238
p16 status	0.061	0.481	0.224	1.034

PPARγ cytoplasm = expression of PPARγ in cytoplasm with IRS ≥ 2, pT = tumor stage, pN = nodal stage, FIGO = Classification of the International Federation of Gynecology and Obstetrics.

**Table 6 diagnostics-11-00491-t006:** Cox regression of clinical-pathological variables regarding overall survival in vulvar carcinoma patients.

Variable	Significance	Hazard Ratio of Exp (B)	Lower 95% Confidence Interval of Exp (B)	Upper 95% Confidence Interval of Exp (B)
PPARγ+/COX-2	0.001	2.615	1.460	4.683
pT	0.613	1.195	0.598	2.388
pN	0.85	1.006	0.999	1.013
Grading	0.012	1.738	1.129	2.676
FIGO	0.098	1.453	0.933	2.263
p16 status	0.002	0.380	0.208	0.694

PPARγ+/COX-2 = expression of PPARγ and/or COX-2 in cytoplasm (IRS), pT = tumor stage, pN = nodal stage, FIGO = Classification of the International Federation of Gynecology and Obstetrics.

## Data Availability

The data presented in this study are available on request from the corresponding author. The data are not publicly available due to ethical issues.

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
