# Peer review of "Combined COX-2/PPARγ Expression as Independent Negative Prognosticator for Vulvar Cancer Patients"

_diagnostics, 2021, doi:10.3390/diagnostics11030491_

Round 1

Reviewer 1 Report

Here author trying to show that COX-2/ PPAR both act as independent negative prognosticator for vulvar cancer patients. However, present manuscript is not of publication quality and require improvement:
1. All the figures quality is very poor. Needs to improve.
2. There are lots of spelling and grammatical mistakes. It needs to improve.
3. Some sontences are not clear such as "Together with the other subtypes PPAR and PPAR / , PPARg". What author is trying to explain?
4. Author should also mention the status of these proteins COX-2/ PPAR in chemoresistant patient.

Author Response

Response to Reviewer 1

First of all, we would like to thank you for reviewing the paper and your interesting comments and remarks. In the following you will find our answers to your questions along with descriptions of what we reworked or added to our updated version of the paper.

Reviewer 1, comment 1

All the figures quality is very poor. Needs to improve.
Reply to comment 1:

The figures have been completely revised again and inserted in improved quality.

Reviewer 1, comment 2

There are lots of spelling and grammatical mistakes. It needs to improve.

Reply to comment 2:

The errors regarding the language in general as well as grammatical or spelling mistakes have been corrected.

Reviewer 1, comment 3

Some sontences are not clear such as "Together with the other subtypes PPAR and PPAR / , PPARg". What author is trying to explain?

Reply to comment 3:

We took the valuable point you brought up so additionally you can find now in the reworked paper version the following part (line 73f):

The nuclear receptor superfamily for steroids, hormones, vitamin D, and retinoid is formed by isoforms like PPARa, PPARß, and PPARγ (peroxisome proliferator activated receptor gamma).

Reviewer 1, comment 4

Author should also mention the status of these proteins COX-2/ PPAR in chemoresistant patient.

Reply to comment 4:

It is definitely an interesting aspect to study COX-2 and PPARγ in tissues taken from chemoresistant patients. Though, in the guideline-based therapies for patients suffering from vulvar carcinoma, surgical therapy and radiation therapy is still in the lead internationally. Chemotherapy has not yet been widely adopted in the treatment of vulvar cancer, and for this reason we unfortunately do not have any patients in the study population who received chemotherapy or who turned out to be chemoresistant.

Reviewer 2 Report

The paper by Ansorge et al, is original and may be interesting but as the authors recognise, the study has many limitations and constraints.

To mention some drawbacks: 

First of all, it is a retrospective study with a high number of patients, 141 slides, but probably not representative of the vulvar cancer at different ages, since the median age of the collective is 70 years, and then, it would seem that it it a later age cancer, therefore, the survival prognosis is not so critical. However, the range is from 20 to 96 years, with half of them younger than 70. This is an indicator of a non homogenous group, and the authors have not clustered the results by age intervals. And this is important when talking about inflammatory markers like COX-2, they increase in expression with age. It cannot have  the same meaning a high expression in a vulvar cancer of a 20 years old women, than in a 90 years old.

On the other hand, quantification of expression of both markers COX-2 and PPAR should have been measured in the surrounding non tumoral tissue to test difference between tumoral tissue and apparently non tumoral tissue.

On the other hand, the quality of staining and contrast of the IHC is very poor and differences are not easy to see.

Only one method: immunostaining of paraffin samples has been used to assess the diagnostic value of PPAR and COX-2. As the authors recognize, this is a limitation and weakness, since other methods like RT-qPCR, and biomarkers in sera of patients would be supportive. 

Therefore this reviewer feels that the paper is not appropriate to be published in the present form.

Author Response

Response to Reviewer 2

First of all, we would like to thank you for reviewing the paper and your interesting comments and remarks. In the following you will find our answers to your questions along with descriptions of what we reworked or added to our updated version of the paper.

Reviewer 2, comment 1

First of all, it is a retrospective study with a high number of patients, 141 slides, but probably not representative of the vulvar cancer at different ages, since the median age of the collective is 70 years, and then, it would seem that it is a later age cancer, therefore, the survival prognosis is not so critical. However, the range is from 20 to 96 years, with half of them younger than 70. This is an indicator of a non homogenous group, and the authors have not clustered the results by age intervals. And this is important when talking about inflammatory markers like COX-2, they increase in expression with age. It cannot have the same meaning a high expression in a vulvar cancer of a 20 years old women, than in a 90 years old.
Reply to comment 1:

The Robert-Koch-Institute Germany reports regarding the epidemiology of vulvar carcinoma a medium age of 73 years, which is manifested in the highest burden of disease within the group of women over 70. So in this case our collective represents an approximation of the frequency of this disease within the age groups. The aim of our study is to find a prognostic factor for all vulvar carcinoma patients. Due to the rarity of this type of carcinoma, especially in comparison to other types e.g. breast cancer, the sample size of our collective is unprecedented in common literature.

Reviewer 2, comment 2

On the other hand, quantification of expression of both markers COX-2 and PPAR should have been measured in the surrounding non tumoral tissue to test difference between tumoral tissue and apparently non tumoral tissue.

Reply to comment 2:

As our study is designed with a retrospective analysis there was no possibility to overview the process of the embedment of the tissue which leads to the fact that in the analyzed collective only tumorous tissue is accessible. Nevertheless it is of great progress to the science of vulvar carcinomas if it would be possible for a future financially supported panel study to embed separately not only tumorous but non-malignant tissue as well. We will take up this point in the planning of upcoming projects.

Reviewer 2, comment 3

On the other hand, the quality of staining and contrast of the IHC is very poor and differences are not easy to see.

Reply to comment 3:

For the updated version of the paper the general quality of the pictures was improved.  Regarding an improved possibility of differentiation we revised our material in an extensive process to enable a reselection for pictures of higher informative value.

Reviewer 2, comment 4

Only one method: immunostaining of paraffin samples has been used to assess the diagnostic value of PPAR and COX-2. As the authors recognize, this is a limitation and weakness, since other methods like RT-qPCR, and biomarkers in sera of patients would be supportive. 

Reply to comment 4:

Immunohistochemistry was applied in our single-method approach due to the fact that this technique of advanced histopathological diagnostic is well established and renowned within the respective field of research. In alignment to our answer under point 2 also this limitation comes along with the retrospective design of the study. Only a highly limited number of patients included in this collective (Period of the premier diagnosis of the patient collective 1990-2008) was alive during the examination of the tissue sections so that there would have been the possibility of applying a scan for cells within serum or primary cells. Though, for a prospective clinical study this aspect would deliver an interesting impulse to the efforts of our research group to establish a prospective biomarker.     

Reviewer 3 Report

The present study presents a high interest for the research community as the number of research papers in vulvar cancer and COX-2 expression is limited, especially in the last ten years. When it comes to PPAR or the combination between COX-2 and PPAR no evidence of there study as biomarkers for vulvar cancer appears in the literature. The research approach is accurate as well as the statistical analysis used. Therefore I support the publication of the paper with a observation. That it would be interesting for the paper to show the expression of both genes in normal vulvar tissue. Did you check their expression/localization through immunhistochemistry staining in normal tissue?

Author Response

Response to Reviewer 3

First of all, we would like to thank you for reviewing the paper and your interesting comments and remarks. In the following you will find our answers to your questions along with descriptions of what we reworked or added to our updated version of the paper.

Reviewer 3, comment 1

Therefore I support the publication of the paper with an observation. That it would be interesting for the paper to show the expression of both genes in normal vulvar tissue. Did you check their expression/localization through immunohistochemistry staining in normal tissue?

Reply to comment 1:

For the presented research in the paper also benign tissue was examined. Thanks to your impulse we could include the referring picture in the updated version of our paper in the supplementary folder which also strengthens our presented argument.

Reviewer 4 Report

Ansorge et al., investigated both the influence of COX2 and PPAR expression in vulvar cancer patients and aim to use it as a prognostic factor. The authors address that future research should aim at linking the two pathways the authors don’t show any pathway link in their manuscript and only limited in their discussion although there are in the literature already some established links which could have been investigated. Please address this point more into depth in your discussion.

What the authors did already address in their discussion section the complete study evolves around one method is based on IRS values however regarding this point there are some questions that need to be addressed.

Could the authors be more descriptive on the IRS and what is stands for and how it works. The authors refer to a German article which isn’t widely available although it is highly cited. But maybe the authors can provide more details? Now it looks like for COX2 IRS of >3 is chosen as a cut-off while for PPAR>=2 is chosen. Could the authors provide more details on the thresholds and if e.g. for COX2 IRS>=2 didn’t correlate to a decreased survival rate?

All the figure and table captions aren’t descriptive but use words as clear, impressive etc. as the those are part of the results of the manuscript the captions need to be more descriptive of the data. Explain what are the blue lines versus the red lines, as those in the graph are hardly readable. Please rewrite all captions. 

Line 279, the authors refer to the paper of Fons et al., at the end of this sentence they write: …in this study. This makes it confusing for the reader. Please omit this.

Could more phosphorylation of MAPK be observed in the slides potentially explaining the increase of cytoplasmic PPAR.

Author Response

Response to Reviewer 4

First of all, we would like to thank you for reviewing the paper and your interesting comments and remarks. In the following you will find our answers to your questions along with descriptions of what we reworked or added to our updated version of the paper.

Reviewer 4, comment 1

The authors address that future research should aim at linking the two pathways the authors don’t show any pathway link in their manuscript and only limited in their discussion although there are in the literature already some established links which could have been investigated. Please address this point more into depth in your discussion.

Reply to comment 1:

We took the valuable point you brought up so additionally you can find now in the reworked paper version the following part (line 236-244):

COX-2 plays a decisive role in carcinogenesis. Acting COX-2 products, prostanoids, appear to be linked to the development and progression of tumor disease. Processes including angiogenesis, invasion, apoptosis inhibition, growth and aggressiveness of the tumor seem to depend strongly on COX-2 and its products.(43-45) Various cross-links to signaling pathways via NF-kB (46), Wnt/ß-catenin (47), PI3K/AKT(48) or activations of anti-apoptotic Bcl-2(49) are established by COX-2. The NF-kB pathway regulates the expression of COX-2. As has already been investigated in several studies, PPARγ in its activated form acts in the nucleus as an inhibitory factor on the transcription factor NF-kB. PPARγ thereby inhibits the expression of COX-2, which can be regarded as an important connection point between the two proteins.(46, 50, 51)

Reviewer 4, comment 2

Could the authors be more descriptive on the IRS and what is stands for and how it works.

Reply to comment 2:

Considering your comments on the immunoreactive score we included a short explanation of the IRS in the updated version of the paper (line 393-398):

Under use of the semi quantitative immunoreactive score (IRS) by Remmele and Stegner (104) tissue patterns were evaluated with the light microscope (Leitz, Wetzlar, Germany). For this purpose the intensity score and the percentage score in the tumor tissue is formed. The intensity score is divided into 0= no, 1= weak, 2= moderate, 3= strong; the percentage score is also categorized: 0= no staining, 1= < 10%, 2= 11% to 50%, 3= 51% to 80%, 4= >81%. IRS score is formed by product of both scores (intensity score x percentage score).

Reviewer 4, comment 3

Could the authors be more descriptive on the IRS and what is stands for and how it works. The authors refer to a German article which isn’t widely available although it is highly cited. But maybe the authors can provide more details?

Reply to comment 3:

We would like to apologise that the paper by Remmele et al., which we cited it in our paper, is of no uncomplicated access. Though, these authors seem to have established the IRS, their paper would be considered as primary literature. 

Reviewer 4, comment 4

Now it looks like for COX2 IRS of >3 is chosen as a cut-off while for PPAR>=2 is chosen. Could the authors provide more details on the thresholds and if e.g. for COX2 IRS>=2 didn’t correlate to a decreased survival rate?

Reply to comment 4:

Our study focuses on a potential prognostic marker for patients suffering from vulvar carcinoma. Within the process of gaining the best possible informative value for both of the proteins, COX-2 and PPARγ, different cut-offs where tested and compared so that the published cut-off values for COX-2 and PPARγ differ for exactly this reason.

Reviewer 4, comment 5

All the figure and table captions aren’t descriptive but use words as clear, impressive etc. as the those are part of the results of the manuscript the captions need to be more descriptive of the data. Explain what are the blue lines versus the red lines, as those in the graph are hardly readable. Please rewrite all captions. 

Reply to comment 5:

Your remarks regarding the figures and their referring captions as well as the more precise explanations of the casted lines in the Kaplan-Meier-curves where taken up for the updated version of the paper.

Reviewer 4, comment 6

Line 279, the authors refer to the paper of Fons et al., at the end of this sentence they write: …in this study. This makes it confusing for the reader. Please omit this.

Reply to comment 6:

The passage of our text in the paper, which refers to a citation of Fons et al., is now expressed in a different way to enable a better understanding for the readers of our paper.

Reviewer 4, comment 7

Could more phosphorylation of MAPK be observed in the slides potentially explaining the increase of cytoplasmic PPAR.

Reply to comment 7:

So far the tissue sections have not been examined for MAPK-activity. Considering your remark we again went into the modelling of PPARγ especially in context of processes of phosphorylation. So as a result of that you will find below the updated version of the referring text passage from the paper (line 280-285):

Other factors influence the activity regulation of PPARγ like ubiquitination or the influence of natural ligands like PGJ2 or external ligands like thiazolidiones (90). Additional influences by post-translational modification via phosphorylation by MAPK as well as activity modulation by traditional herbal medicine plants like V.trifolia demonstrate the diversity of possible modulatory pathways of the activation and expression pattern of PPARγ.(91, 92)

Round 2

Reviewer 2 Report

Since most of the concerns were not addressed but just justified, I don´t consider that the manuscript has improved enough to be published.

Author Response

First of all, we want to thank you for your note that made it possible for us to again improve our paper.

We included our answers based on the questions and remarks brought up by reviewer 2 as we were asked to in the discussion segment of our study as limitations. In the points below you can find again the comments made by reviewer 2 as well as the citation of the referring part of our updated “limitations of the study” segment in our paper.

Reviewer 2, comment 1

First of all, it is a retrospective study with a high number of patients, 141 slides, but probably not representative of the vulvar cancer at different ages, since the median age of the collective is 70 years, and then, it would seem that it is a later age cancer, therefore, the survival prognosis is not so critical. However, the range is from 20 to 96 years, with half of them younger than 70. This is an indicator of a non homogenous group, and the authors have not clustered the results by age intervals. And this is important when talking about inflammatory markers like COX-2, they increase in expression with age. It cannot have the same meaning a high expression in a vulvar cancer of a 20 years old women, than in a 90 years old.

Reply to comment 1:

“In 2016, the Robert-Koch-Institute Germany reports regarding the epidemiology of vulvar carcinoma a medium age of 73 years, which is manifested in the highest burden of disease within the group of women over 70. (108) In this case our collective represents an approximation of the frequency of this disease within the age groups. The aim of our study is to find a prognostic factor for all vulvar carcinoma patients. Due to the rarity of this type of carcinoma, especially in comparison to other types e.g. breast cancer, the sample size of our collective is unprecedented in common literature.” {line 459-465 in manuscript}

Reviewer 2, comment 2

On the other hand, quantification of expression of both markers COX-2 and PPAR should have been measured in the surrounding non tumoral tissue to test difference between tumoral tissue and apparently non tumoral tissue.

Reply to comment 2:

“There was no possibility to overview the process of the embedment of the tissue which leads to the fact that in the analyzed collective only tumorous tissue is accessible. Vulvar tissue patterns of patients with non-malignant vulvar diseases were tested by immunohistochemical expression level for COX-2 and PPARg for clarifying differences between malignant and non-malignant tissue expression levels (Suppl. 1,2).“ {line 436-440 in manuscript}

Reviewer 2, comment 3

On the other hand, the quality of staining and contrast of the IHC is very poor and differences are not easy to see.

Reply to comment 3:

For the updated version of the paper the general quality of the pictures was improved. Regarding an improved possibility of differentiation we revised our material in an extensive process to enable a reselection for pictures of higher informative value. {Figures 1 and Figure 3 in manuscript}

Reviewer 2, comment 4

Only one method: immunostaining of paraffin samples has been used to assess the diagnostic value of PPAR and COX-2. As the authors recognize, this is a limitation and weakness, since other methods like RT-qPCR, and biomarkers in sera of patients would be supportive.

Reply to comment 4:

“Immunohistochemistry was applied in our single-method approach due to the fact that this technique of advanced histopathological diagnostic is well established and renowned within the respective field of research. This limitation comes along with the retrospective design of the study. Only a highly limited number of patients included in this collective was alive during the examination of the tissue sections so that there would have been the possibility of applying a scan for cells within serum or primary cells.” {line 428-433 in manuscript}
